# Incidence and Practice of Early Prone Positioning in Invasively Ventilated COVID-19 Patients—Insights from the PRoVENT-COVID Observational Study

**DOI:** 10.3390/jcm10204783

**Published:** 2021-10-19

**Authors:** Willemke Stilma, David M. P. van Meenen, Christel M. A. Valk, Hendrik de Bruin, Frederique Paulus, Ary Serpa Neto, Marcus J. Schultz

**Affiliations:** 1Department of Intensive Care, Amsterdam UMC, Location ‘AMC’, 1105 AZ Amsterdam, The Netherlands; d.m.vanmeenen@amsterdamumc.nl (D.M.P.v.M.); c.m.valk@amsterdamumc.nl (C.M.A.V.); h.debruin2@amsterdamumc.nl (H.d.B.); f.paulus@amsterdamumc.nl (F.P.); marcus.j.schultz@gmail.com (M.J.S.); 2Center of Expertise Urban Vitality, Faculty of Health, Amsterdam University of Applied Sciences, 1105 BD Amsterdam, The Netherlands; 3Australian and New Zealand Intensive Care Research Centre (ANZIC-RC), Monash University, Melbourne 3004, Australia; ary.serpaneto@monash.edu; 4Department of Critical Care Medicine, Hospital Israelita Albert Einstein, São Paulo 05652-900, Brazil; 5Mahidol–Oxford Research Unit (MORU), Faculty of Tropical Medicine, Mahidol University, Bangkok 10400, Thailand; 6Nuffield Department of Medicine, University of Oxford, Oxford OX3 7BN, UK

**Keywords:** coronavirus disease 2019, COVID-19, ARDS, prone positioning, intensive care, critical care, artificial ventilation, mortality

## Abstract

We describe the incidence and practice of prone positioning and determined the association of use of prone positioning with outcomes in invasively ventilated patients with acute respiratory distress syndrome (ARDS) due to coronavirus disease 2019 (COVID-19) in a national, multicenter observational study, performed at 22 intensive care units in the Netherlands. Patients were categorized into 4 groups, based on indication for and actual use of prone positioning. The primary outcome was 28-day mortality. Secondary endpoints were 90-day mortality, and ICU and hospital length of stay. In 734 patients, prone positioning was indicated in 60%—the incidence of prone positioning was higher in patients with an indication than in patients without an indication for prone positioning (77 vs. 48%, *p* = 0.001). Patients were left in the prone position for median 15.0 (10.5–21.0) hours per full calendar day—the duration was longer in patients with an indication than in patients without an indication for prone positioning (16.0 (11.0–23.0) vs. 14.0 (10.0–19.0) hours, *p* < 0.001). Ventilator settings and ventilation parameters were not different between the four groups, except for FiO_2_ which was higher in patients having an indication for and actually receiving prone positioning. Our data showed no difference in mortality at day 28 between the 4 groups (HR no indication, no prone vs. no indication, prone vs. indication, no prone vs. indication, prone: 1.05 (0.76–1.45) vs. 0.88 (0.62–1.26) vs. 1.15 (0.80–1.54) vs. 0.96 (0.73–1.26) (*p* = 0.08)). Factors associated with the use of prone positioning were ARDS severity and FiO_2_. The findings of this study are that prone positioning is often used in COVID-19 patients, even in patients that have no indication for this intervention. Sessions of prone positioning lasted long. Use of prone positioning may affect outcomes.

## 1. Introduction

Patients with acute respiratory distress syndrome (ARDS) have been shown to benefit from early prone positioning if hypoxemia is severe and refractory through an increase in the fraction of inspired oxygen (FiO_2_) > 60% and higher positive end-expiratory pressure (PEEP) [1,2]. Especially patients with focal consolidations could profit from this intervention [3], as higher PEEP may be ineffective and could even cause overdistension. Before the coronavirus disease 2019 (COVID-19) pandemic, prone positioning remained remarkably underused [4,5].

Invasively ventilated patients with ARDS due to COVID-19 often have an indication for prone positioning. Indeed, these patients often have severe hypoxemia. Additionally, consolidation may behave as focal lesions [6,7], which is another reason to apply prone positioning early after the start of invasive ventilation [8]. Last but not least, hypoxemia could also be a consequence of pulmonary embolism, for which higher PEEP is not helpful. Several recent reports in COVID-19 patients have shown frequent use of prone positioning, but with a remarkable variance in incidence and practice [6,8,9,10,11].

The purpose of this current analysis of a national multicenter study of COVID-19 patients admitted to the intensive care unit (ICU) for invasive ventilation early in the pandemic, named ‘PRactice of VENTilation in COVID-19′ (PRoVENT-COVID) [12], was to study the incidence and practice of prone positioning in this cohort. We tested the hypothesis that prone positioning improves the outcome of COVID-19 patients. We also wished to determine what factors were associated with its use.

## 2. Methods

### 2.1. Study Design

The PRoVENT-COVID study is an observational cohort study undertaken at 22 ICUs from 1 March 2020 until 1 June 2020 in the Netherlands—in this study, we enrolled ~40% of all patients that needed invasive ventilation during the first wave of the national outbreak [13]. The study protocol [12] and the statistical analysis plan for the current analysis were prepublished [14].

### 2.2. Ethics

The study protocol was approved by the ethics committee in Amsterdam UMC, Amsterdam, the Netherlands (registration number W20_157 # 20.171); the need for individual patient informed consent was waived due to the observational nature of the study.

### 2.3. Study Registration

The study was registered at clinicaltrials.gov on 15 April 2020 with study identifier NCT04346342.

### 2.4. Inclusion and Exclusion Criteria

Consecutive patients were enrolled in the PRoVENT-COVID study if (1) age was >18 years; (2) they were admitted to one of the participating ICUs; and (3) they had received invasive ventilation for respiratory failure related to COVID-19 that was confirmed by a reverse transcriptase–polymerase chain reaction for SARS-CoV-2. The PRoVENT-COVID study had no exclusion criteria. For the current analysis, we excluded patients who were transferred from or to another ICU during the first days of invasive ventilation, as it could be that prone positioning was delayed because of an imminent transport and also because data on the use of prone positioning could not be assessed in non-participating centers.

### 2.5. Collected Data, and Patient Classification

We collected demographic data, including disease severities and the medical history at baseline. ARDS severity was scored as mild, moderate or severe, in accordance with the current definition for ARDS [15]. Ventilator settings and ventilation parameters were collected every 8 h, and use and timing of prone positioning and use of neuromuscular blocking agents (NMBA) were collected in the first 4 calendar days of invasive ventilation. Chest X-rays and lung CT-scans were made at initiation of invasive ventilation. The X-rays were coded in quadrants and the CT-scans as a %. This was based on the interpretation of trained data collectors: all data collectors had a medical background and had received additional training regarding the chest X-rays and lung CT-scan assessment before the start of data collection. Follow-up was complete up to day 90, and included timing of liberation from invasive ventilation, ICU and hospital discharge, and life status at ICU and hospital discharge, and at day 28 and day 90.

Patients were categorized into 4 groups based on indication for (yes or no) and the use of (yes or no) early prone positioning. A patient was labeled to have an indication for prone positioning if PaO_2_/FiO_2_ ratio < 150 mmHg, at PEEP of ≥5 cm H_2_O and FiO_2_ ≥ 0.6 [1] for at least 2 consecutive time points within the first 32 h after the start of invasive ventilation.

### 2.6. Study Endpoints

The primary endpoint of this analysis was 28-day mortality. Secondary outcomes were 90-day mortality, and ICU- and hospital length of stay (LOS).

### 2.7. Statistical Analyses

We did not perform a formal sample size calculation; instead, the number of available patients served as the sample size. The day of intubation, which in theory could last from 1 min to 23 h and 59 min, was named ‘day 0’. Successive days were named ‘day 1’, ‘day 2’ and ‘day 3’.

Categorical patient variables are presented as numbers and percentages, and continuous data as medians with interquartile ranges. With regard to the primary endpoint, there were no missing data. The amount of missing data of other variables was low, <5%. Incidence of prone positioning is expressed as numbers and percentages variables. Timing and duration of prone positioning are expressed in the number of hours from the start of invasive ventilation, and the total number of hours per full calendar day. To assess differences among the 4 groups a chi-squared test and Kruskal–Wallis test were used where appropriate.

Ventilatory variables and parameters over the first 4 calendar days were compared using a Kruskal–Wallis test and were presented in cumulative distribution plots and line graphs displaying the 4 groups of interest. For each day, ventilatory variables at the moment of the worst PaO_2_/FiO_2_ for that day were used, assuming these were collected at the moment the patient was in a supine position.

Hazard ratios (HRs) for 28-day and 90-day mortality were compared between the 4 groups using a (shared-frailty) Cox proportional hazard model, with the center as frailty. HRs for ICU length of stay, hospital length of stay and ventilator free days, were compared using a competing risk analysis with the center as a random effect. Kaplan–Meier curves were constructed for all outcomes of interest. Predefined variables assessed for the final models were severity of ARDS [10], PEEP, FiO_2_, body mass index (BMI), use of NMBAs and tidal volume per predicted body weight. If these variables had a *p* < 0.20 in the univariable model, they were included in the multivariable model. Covariates used for the final model were the variables with a *p* < 0.05 in the multivariable model; the covariates used in the univariable, and multivariable models are reported in Appendix A. This analysis was repeated to compare patients having an indication for and receiving prone positioning and patients having an indication for but not receiving prone positioning.

An adjusted mixed-effect model with the center as a random effect was used to determine which factors had an association with use of prone positioning. Variables included in this model were severity of ARDS [16], PEEP, FiO_2_, body mass index (BMI) and hypercapnia.

As a posthoc analysis, a time-dependent Cox regression analysis was performed. All models were checked for collinearity. All analyses were conducted in R v.4.0.3 (R Foundation for Statistical Computing: Vienna, Austria) [17] and a *p* < 0.05 was considered significant.

## 3. Results

### 3.1. Patients Enrolled

Between 1 March and 1 June 2020, 22 ICUs were invited and accepted participation in the PRoVENT-COVID study. Of 1122 enrolled patients, 734 patients were eligible for the current analysis. The main reason for exclusion was an early transfer from or to a non-participating hospital (Figure 1). At the start of ventilation, patients that were placed in prone positioning had higher severities of ARDS, and PaO_2_/FiO_2_ < 150 was more frequent in patients that had an indication for prone positioning (Table 1). Additionally, in the group without an indication for prone positioning, the severity of ARDS and the number of patients with a PaO_2_/FiO_2_ ≤ 150 mm Hg was higher in the group that received prone positioning than the group that was not placed in the prone position (Table 1). NMBAs were used more often in patients having an indication than in patients not having an indication for prone positioning (60% vs. 52%).

### 3.2. Incidence of Prone Positioning

Prone positioning was used in 438/734 (60%) patients. Incidence of prone positioning was higher in patients having an indication than in patients not having an indication for prone positioning (77% vs. 48%; *p* < 0.001). For patients who were placed in the prone position, the median first day of proning was day 0 (0–1) and received prone positioning over a median of 3 (2–3) days; this was similar in patients with and without an indication. Prone positioning lasted a median of 16.0 (11.0–23.0) hours per full calendar day in patients having an indication, and 14.0 (10.0–19.0) hours in patients without an indication (*p* < 0.001) (Table 2 and Figure 2).

### 3.3. Ventilation Characteristics in the First 4 Calendar Days of Ventilation

At the start of ventilation, peak airway pressure, driving pressure, compliance, respiratory rate, FiO_2_, PaO_2,_ and SaO_2_/FiO_2_ and PaO_2_/FiO_2_, and mechanical power were different between the 4 groups (Table 3). In the group without an indication for prone positioning, invasive ventilation differed with regard to the peak and driving pressure. Both were higher in the patients that received prone positioning (Table 3). Driving pressure, compliance, PaO_2_/FiO_2_, FiO_2_ and PaCO_2_ remained different between groups on successive days. Tidal volume was not different between groups on any day of collection of these data. PEEP was only different on day 2 and day 3; PaO_2_ was different on day 0 and day 1 (Figure 3, Figure 4 and Appendix A).

### 3.4. Patient Outcomes

Mortality at day 28 was lowest in patients with no indication for prone positioning—28.6% vs. 31.3% in patients that were not placed in prone position vs. patients that were placed in the prone position. Mortality at day 28 was highest in patients with an indication for prone positioning—41.3% vs. 34.1% in patients that were not placed in prone position vs. patients that were placed in the prone position. Differences between the four groups, though, did not reach statistical significance (*p* = 0.244). Differences in mortality at day 90 between groups followed a similar pattern (*p =* 0.100).

ICU length of stay in patients that survived till ICU discharge was lower in patients that had no indication for prone positioning—a median of 16 (10–25) days vs. 19 (12–33) days, in patients that were not placed in a prone position vs. patients that were placed in a prone position. ICU length of stay in patients that survived till ICU discharge was higher in patients that had an indication for prone positioning—a median of 22 (12–30) days vs. 21 (14–34) days in patients that were not placed in a prone position vs. patients that were placed in a prone position.

Hospital length of stay in patients that survived till hospital discharge was lower in patients with no indication for prone positioning—a median of 28 (20–40) days vs. 31 (22–) days in patients that were not placed in the prone position vs. patients that were placed in the prone position. Hospital length of stay in patients that survived till hospital discharge was higher in patients with an indication for prone positioning—a median of 31 (21–44) days vs. 35 (24–50) days in patients that were not placed in the prone position vs. patients that were placed in the prone position.

Ventilator free days at day 28 were higher in patients with no indication for prone positioning—a median of 7.0 (0.0–17.5) days vs. 1.0 (0.0–17.00) days, in patients that were not placed in the prone position vs. patients that were placed in the prone position. Ventilator free days at day 28 were low in patients with an indication for prone positioning—a median of 0.0 (0.0–10.0) days vs. 0.0 (0.0–14.0) days, in patients that were not placed in a prone position vs. patients that were placed in a prone position.

Adjusted HRs were different between groups for mortality at day 90, ICU length of stay and hospital length of stay, but not for mortality at day 28 (Figure 5 and Appendix A).

### 3.5. Factors That Have an Association with Use of Prone Positioning

ARDS severity and FiO_2_ were the only factors that were independently associated with the actual use of prone positioning (Appendix A).

### 3.6. Post Hoc Analysis

The time-dependent Cox regression analysis did not change the findings (Appendix A).

## 4. Discussion

Here, we describe the practice and outcome of prone positioning in patients with ARDS due to COVID-19 that received invasive ventilation in the first 3 months of the national outbreak in the Netherlands. The incidence of prone positioning was high, also in patients not having an indication for this intervention. Sessions of prone positioning were long and lasted longer in patients with an indication. ARDS severity and FiO_2_ predicted the use of prone positioning.

Our study confirms the high incidence of prone positioning in invasively ventilated COVID-19 patients, as found in other observational studies [6,9,11,18]. Studies from before the COVID-19 pandemic showed a remarkable underuse of this intervention in patients with ARDS—in the LUNG SAFE study in 2014, overall use was 7.9%, and 16.3% in patients with severe ARDS [19]; in the APRONET study in 2016, overall use was 13.7%, and 32.9% of patients with severe ARDS [5]. There are several reasons why prone positioning is used more often in COVID-19 patients. It could simply be that the increase of use has continued after LUNG SAFE and APRONET—the implementation of interventions with proven benefits can take many years, also in the ICU setting [20]. It could also be that the poor results of randomized clinical trials that tested alternative ways to improve outcomes, like higher PEEP and recruitment maneuvers, have had a positive effect on the use of prone positioning. Last but not least, it could be that COVID-19 ARDS presents as a form of lung injury that may respond better to prone positioning than other forms of ARDS [3]. Indeed, the findings of one randomized clinical trial suggest that prone positioning may be better than higher PEEP in ARDS patients with non-recruitable lung lesions, which may be typical in COVID-19 ARDS, at least at the initiation of invasive ventilation [3,21].

The high incidence of prone positioning was notable in patients not having an indication for this intervention. This may also have been the case in other cohorts, as the reported overall PaO_2_/FiO_2_ ratio in other studies was comparable to that in our cohort [5,8,14], suggesting a similar distribution of ARDS severities and with that a comparable rate of indication for prone positioning. In addition, some of the patients without an indication for prone positioning were actually placed in the prone position. This group had median lower PaO_2_/FiO_2_ ratios, which could be seen as an indication to initiate prone positioning by the clinician. Whether the results of the chest X-rays and lung CT-scans were an indication for the clinician to initiate prone positioning could not be collected. Therefore, we could not comment on or analyze the relationship between imaging and the indication for prone positioning in this cohort.

In our data, there was no difference in mortality at day 28 between groups, but there was a difference in mortality at day 90. Duration of invasive ventilation is remarkably high in COVID-19 patients, and so is LOS in ICU in these patients [11,12,13,14,15,16,17,18,19,20,21,22,23,24,25,26,27,28,29,30]. This can explain why 28-mortality was not different between the groups, while 90-day mortality was. On the one hand, it could suggest the benefit of this intervention in ARDS due to COVID-19, in line with the findings of the seminal study in patients with ARDS not related to COVID-19 in France [1]. This finding is also in line with the results of one meta-analysis of studies in invasively ventilated COVID-19 patients [23]. On the other hand, it could be that this intervention was foregone in patients with a poor outcome, or in patients with treatment directives. Information on this was not collected in the PRoVENT-COVID study. This explanation, however, seems less likely as there were no differences in any baseline characteristic. The use of NMBA during prone positioning was remarkably lower than in previous studies in the pre-COVID era, in which the incidences were as high as 72% [23] to 88% [11]. It should be noted, though, that recommendations regarding the use of NMBA in ARDS patients [24] may have become obsolete after the publication of the more recent ROSE trial [25]. Additionally, recommendations for the use of NMBA, as well as the actual use of NMBA during prone positioning, may depend on local practices, and maybe even on the experiences of the healthcare workers that had to take care of patients in the overwhelming first wave of the COVID-19 pandemic. Notably, a recent study in COVID-19 patients showed NMBA use to be associated with a higher risk of and a longer duration of ventilation and longer ICU LOS, even after propensity matching [26].

A recent study in patients with ARDS due to COVID-19 showed that sustained improvements in oxygenation in response to the first prone positioning session are associated with better outcomes [27]. Recently, the recruitment-to-inflation ratio was suggested as a bedside tool to identify patients that have a high chance of responding well to lung recruitment maneuvers [28]. Unfortunately, we were unable to separate patients based on these approaches, due to the way data were collected.

Prone positioning could come with procedure-related adverse effects [5,29]. We did not collect these data. It could be hypothesized that the incidence of adverse events during a pandemic is high due to the stressful and demanding situation, with increased workloads and the lack of experienced staff. However, when dedicated prone position teams are present, as was often the case in the centers in the Netherlands early in the pandemic, the rate of procedure-related adverse events could also be low [30].

The only two factors that had an association with the actual use of prone positioning were ARDS severity and FiO_2_. This is in line with an earlier observation from before the COVID-19 pandemic [5]. In that study, the major reason for not placing a patient in the prone position was that clinicians deemed hypoxemia not being severe enough. In the current analysis, a PaO_2_/FiO_2_ ratio < 150 mmHg at two successive observations was used as a cutoff for the indication for prone positioning. This is more strict than in the previous study.

This study has strengths. First, the data were collected in a short time frame during which general care for COVID-19 patients did not change. Second, the study was designed to minimize bias by strictly adhering to a predefined statistical analysis plan. Third, the study involved one-third of all COVID-19 ARDS patients receiving invasive ventilation in the first months of the national outbreak in the Netherlands, and patients were enrolled in 22 ICUs from university-affiliated hospitals, teaching hospitals and non-teaching hospitals, contributing to its generalizability.

PRoVENT-COVID also has limitations. As in any observational study, the knowledge that care data were being captured could have interfered with practice—for instance, doctors and nurses in participating centers could have been keener to use prone positioning. In line with the study design, the use of late prone positioning, i.e., after the first 4 days of invasive ventilation, was not collected. This means that we could not report on the associations of late prone positioning, if that happened, with outcomes. Additionally, it should be realized that reasons to exclude patients from prone positioning, such as recent tracheal surgery or sternotomy, pregnancy or presence of wounds or burns, were not collected. It is conceivable, though, that these contra-indications were barely present in this cohort. Nevertheless, this may have introduced misclassification bias. The selection of ICUs was based on the personal contacts between steering committee members and ICUs that participated in recent research projects of ventilation, which could have resulted in an over-representation of units with more experience in prone positioning, and therefore a higher incidence. Similar to other epidemiological studies, access to patients’ data was restricted to data collectors who were granted access only to patients that were labeled eligible for participation by the local doctors—thus, we could not control whether all COVID-19 patients receiving invasive ventilation in participating ICUs were enrolled. Lastly, the national character of PRoVENT-COVID may make these results not representative for other countries.

The findings of this study extend our understanding of the incidence and practice of prone positioning in patients with ARDS due to COVID-19, and the association of this intervention with outcomes. Our findings may have important suggestions for clinical management.

## 5. Conclusions

In this national cohort of patients with ARDS due to COVID-19, prone positioning was frequently used, even in patients that did not have an indication for this intervention. Prone positioning may improve the outcome of invasively ventilated patients with an indication for this intervention. Factors that had an association with its use were ARDS severity and set FiO_2_.

## Figures and Tables

**Figure 1 jcm-10-04783-f001:**
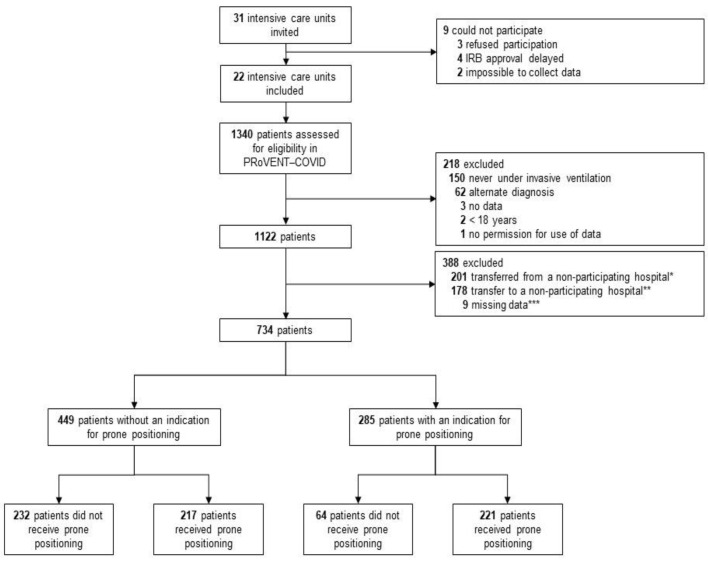
Flow chart of patient inclusion.

**Figure 2 jcm-10-04783-f002:**
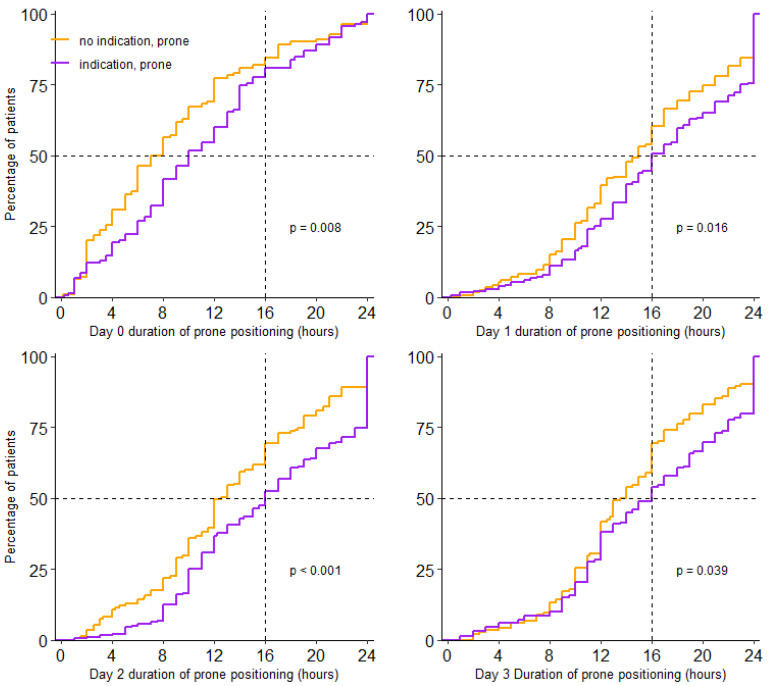
Cumulative distribution duration of prone positioning per day. Duration of prone positioning session for each patient on day 0 to day 3.

**Figure 3 jcm-10-04783-f003:**
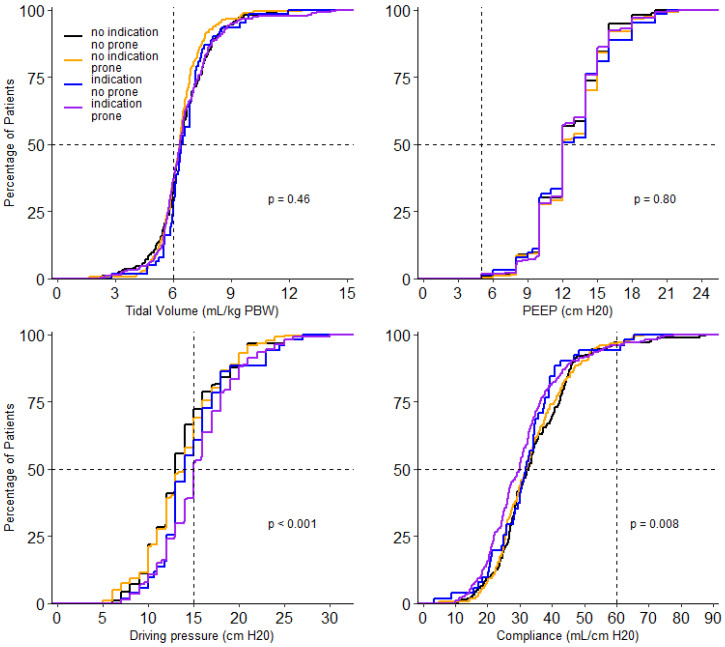
Cumulative distribution of ventilatory parameters on day 0. Levels of tidal volume, PEEP, driving pressure and compliance for each patient on day 0.

**Figure 4 jcm-10-04783-f004:**
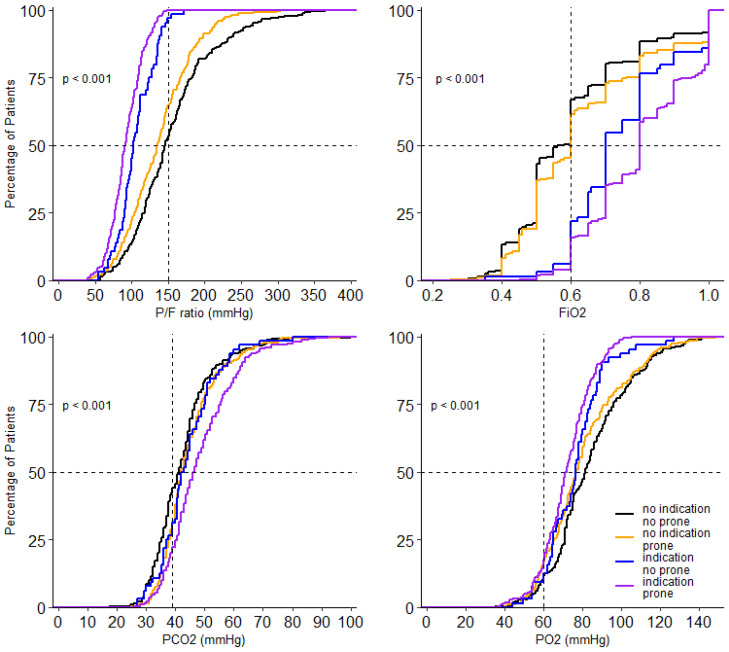
Cumulative distribution of parameters of gas exchange on day 0. Levels of P/F ratio FiO_2_, PO_2_ and PCO_2_ for each patient on day 0.

**Figure 5 jcm-10-04783-f005:**
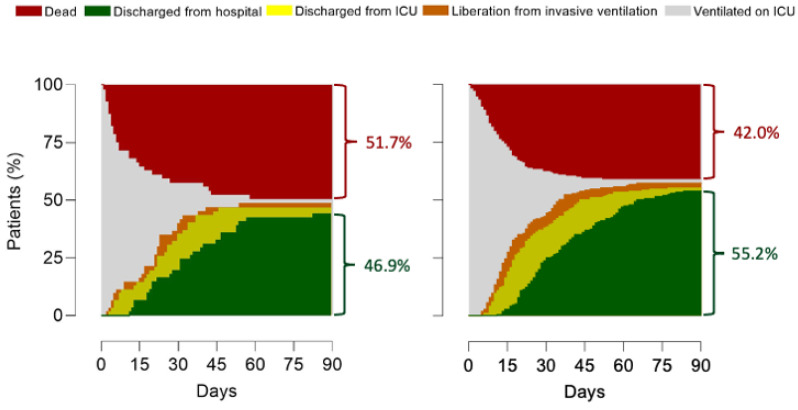
Outcomes. Patient outcomes for the groups of patients with an indication for prone positioning, on the left panel patients are displayed that did not receive prone positioning; on the right panel patients are displayed that did receive prone positioning. HR’s for outcomes were (no indication, no prone vs. no indication, prone vs. indication, no prone vs. indication, prone). 28-day mortality: 1.05 (0.76–1.45) vs. 0.88 (0.62–1.26) vs. 1.15 (0.80–1.54) vs. 0.96 (0.73–1.26) (*p =* 0.08); 90-day mortality: 0.93 (0.67–1.27) vs. 0.89 (0.64–1.24) vs. 1.19 (0.88–1.62) vs. 0.99 (0.76–1.28) (*p =* 0.02); ICU discharge: 1.28 (1.02–1.61) vs. 1.03 (0.80–1.33) vs. 0.88 (0.69–1.12) vs. 0.89 (0.74–1.08) (*p* = 0.02); Hospital discharge: 1.25 (0.99–1.58) vs. 1.07 (0.83–1.39) vs. 0.88 (0.69–1.13) vs. (0.89 (0.73–1.08) (*p* = 0.01); HR’s for outcomes in the groups with an indication were (indication, no prone vs. indication, prone); 28-day mortality: 1.30 (0.82–2.07 vs. 0.76 (0.48–1.21) (*p* = 0.25); 90-day mortality: 1.41 (0.93–2.14) vs. 0.70 (0.46–1.07) (*p* = 0.10); ICU discharge: 0.77 (0.52–1.14) vs. 1.29 (0.87–1.91) (*p* = 0.93); Hospital discharge: 0.78 (0.52–1.18) vs. 1.26 (0.84–1.90) (*p* = 0.70).

**Table 1 jcm-10-04783-t001:** Baseline Characteristics.

	Without an Indication for Prone Positioning			With an Indication for Prone Positioning		
	No Prone Positioning (*N* = 232)	Prone Positioning (*N* = 217)	*p* Value	No Prone Positioning (*N* = 64)	Prone Positioning (*N* = 221)	*p* Value
Age, years (median, IQR)	64.2 (11.4)	65.0 (10.3)	0.430	66.6 (9.1)	62.6 (11.2)	0.008
Male gender, *N* (%)	171 (73.7)	164 (73.2)	0.916	46 (71.9)	154 (69.1)	0.758
BMI, kg/m^2^ (median, IQR)	28.5 (7.8)	28.1 (4.0)	0.490	29.2 (6.3)	29.3 (5.3)	0.841
Chest CT performed, *N* (%)	93 (40.1)	82 (36.6)	0.500	17 (26.6)	92 (41.3)	0.040
Affected lung parenchyma (%)	5 (5.3)	2 (2.4)	0.212	0 (0.0)	2 (2.2)	0.967
≤25%	28 (29.8)	28 (34.1)		6 (35.3)	27 (29.0)	
50%	34 (36.2)	19 (23.2)		5 (29.4)	30 (32.3)	
75%	21 (22.3)	28 (34.1)		5 (29.4)	29 (31.2)	
100%	6 (6.4)	5 (6.1)		1 (5.9)	5 (5.4)	
Chest X-ray performed, *N* (%)	127 (91.4)	135 (93.8)	0.501	44 (93.6)	113 (85.6)	0.199
Number of quadrants affected (%)			0.760			0.790
1	10 (7.9)	10 (7.4)		2 (4.5)	8 (7.0)	
2	27 (21.4)	36 (26.7)		9 (20.5)	27 (23.7)	
3	31 (24.6)	34 (25.2)		16 (36.4)	32 (28.1)	
4	58 (46.0)	55 (40.7)		17 (38.6)	47 (41.2)	
Pneumothorax, *N* (%)	0 (0.0)	0 (0.0)	1.000	0 (0.0)	1 (16.7)	1.000
**Severity of illness**						
SAPS II (median, IQR)	35.7 (11.7)	36.9 (12.8)	0.562	35.9 (16.1)	37.1 (12.8)	0.726
APACHE II (median, IQR)	16.6 (11.3)	18.9 (8.5)	0.230	16.1 (8.0)	19.6 (9.3)	0.295
APACHE IV (median, IQR)	55.2 (20.5)	59.6 (22.9)	0.197	67.2 (25.0)	58.6 (20.6)	0.086
SOFA (median, IQR)	8.1 (3.2)	9.1 (4.4)	0.066	8.6 (3.2)	8.1 (3.8)	0.512
Severity class ARDS (%)			0.002			0.607
Mild	82 (36.3)	51 (23.5)		0 (0.0)	2 (0.9)	
Moderate	142 (62.8)	157 (72.4)		49 (76.6)	154 (69.4)	
Severe	2 (0.9)	9 (4.1)		15 (23.4)	66 (29.7)	
PaO_2_/FiO_2_ ≤ 150 mm Hg, *N* (%)	90 (38.8)	104 (47.9)	0.024	56 (87.5)	189 (85.5)	0.600
**Medical history**						
None	57 (24.6)	46 (20.5)	0.315	14 (21.9)	60 (26.9)	0.517
Hypertension, *N* (%)	75 (32.3)	75 (33.5)	0.842	26 (40.6)	73 (32.7)	0.296
Heart failure, *N* (%)	13 (5.6)	12 (5.4)	1.000	4 (6.2)	7 (3.1)	0.271
Diabetes, *N* (%)	43 (18.5)	57 (25.4)	0.089	12 (18.8)	52 (23.3)	0.499
Chronic kidney disease (%)	9 (3.9)	9 (4.0)	1.000	2 (3.1)	7 (3.1)	1.000
Baseline creatinine, µmol/L (median, IQR)	81.1 (37.2)	96.2 (79.7)	0.021	93.2 (48.1)	82.5 (39.8)	0.121
Liver cirrhosis, *N* (%)	1 (0.4)	0 (0.0)	1.000	1 (1.6)	0 (0.0)	0.223
COPD, *N* (%)	16 (6.9)	17 (7.6)	0.857	5 (7.8)	24 (10.8)	0.640
Active hematological neoplasia, *N* (%)	5 (2.2)	4 (1.8)	1.000	2 (3.1)	0 (0.0)	0.049
Active solid neoplasia, *N* (%)	6 (2.6)	9 (4.0)	0.440	0 (0.0)	5 (2.2)	0.590
Neuromuscular disease, *N* (%)	1 (0.4)	0 (0.0)	1.000	0 (0.0)	5 (2.2)	0.590
Immunosuppression, *N* (%)	10 (4.3)	5 (2.2)	0.295	1 (1.6)	5 (2.2)	1.000
**Home medication**						
Systemic steroids, *N* (%)	13 (5.6)	9 (4.0)	0.515	3 (4.7)	5 (2.2)	0.383
Inhalation steroids, *N* (%)	26 (11.2)	24 (10.7)	0.882	7 (10.9)	29 (13.0)	0.831
ACE inhibitor, *N* (%)	44 (19.0)	48 (21.4)	0.560	7 (10.9)	38 (17.0)	0.329
Angiotensin II receptor blocker, *N* (%)	22 (9.5)	29 (12.9)	0.298	9 (14.1)	21 (9.4)	0.352
Beta blocker, *N* (%)	44 (19.0)	43 (19.2)	1.000	18 (28.1)	43 (19.3)	0.165
Insulin, *N* (%)	17 (7.3)	21 (9.4)	0.499	1 (1.6)	9 (4.0)	0.467
Metformin, *N* (%)	32 (13.8)	44 (19.6)	0.103	7 (10.9)	36 (16.1)	0.426
Statin, *N* (%)	63 (27.2)	76 (33.9)	0.127	18 (28.1)	76 (34.1)	0.450
Calcium channel blockers, *N* (%)	31 (13.4)	49 (21.9)	0.019	15 (23.4)	39 (17.5)	0.281

Abbreviations: ACE inhibitor, Angiotensin-converting enzyme inhibitor; APACHE, Acute Physiology and Chronic Health Evaluation; ARDS, Acute respiratory distress syndrome; COPD, Chronic obstructive pulmonary disease; IQR, interquartile range; SAPS, Simplified Acute Physiology Score; SOFA, Sequential Organ Failure Assessment; BMI, body mass index.

**Table 2 jcm-10-04783-t002:** Duration of Prone Positioning.

	No Indication for Prone Positioning (*N* = 217)	Indication for Prone Positioning (*N* = 221)	*p* Value
Day 0			
*n/N* (%)	110/217 (51)	130/221 (59)	
absolute time in a prone position, hours (median, IQR)	7.5 (3.6–12.0)	10.0 (6.0–14.4)	0.008
relative time in a prone position, % of total hours *	75% (65–75)	100% (80–100)	<0.001
Day 1			
*n/N* (%)	167/217 (77)	180/221 (81)	
absolute time in a prone position, hours (median, IQR)	15 (10.0–20.5)	16 (11.9–23.1)	0.016
relative time in a prone position, % of total hours	63% (42–85)	67% (50–96)	0.016
Day 2			
*n/N* (%)	147/217 (68)	175/221 (79)	
absolute time in a prone position, hours (median, IQR)	12.5 (9.0–18.8)	16 (10.5–23.5)	<0.001
relative time in a prone position, % of total hours *	52% (38–78)	67% (44–98)	<0.001
Day 3			
*n/N* (%)	143/217 (66)	152/221 (66)	
absolute time in a prone position, hours (median, IQR)	13.8 (10.3–18.0)	16.0 (11.0–22.0)	0.039
relative time in a prone position, % of total hours *	58% (43–75)	67% (46–92)	0.039
Total			
Duration of prone positioning per full calendar day (median, IQR)	14.0 (10.0–19.0)	16.0 (11.0–23.0)	<0.001

* Calendar day 0 could last from 0 to 24 h; in patients with no indication day 0 had 10.0 (5.5–16.1) hours, in patients with an indication day 0 had 10.3 (6.0–18.1) hours.

**Table 3 jcm-10-04783-t003:** Ventilatory Characteristics at Start of Ventilation.

	Without an Indication for Prone Positioning			With an Indication for Prone Positioning		
	No Prone Positioning (*N* = 232)	Prone Positioning (*N* = 217)	*p* Value	No Prone Positioning (*N* = 64)	Prone Positioning (*N* = 221)	*p* Value
**Mode of ventilation**	232	217		64	221	
Volume Control	23 (10.0)	51 (22.8)		6 (9.4)	32 (14.5)	
Pressure Control	127 (55.5)	116 (51.8)		33 (51.6)	131 (59.3)	
Pressure Support	7 (3.1)	4 (1.8)		0 (0.0)	8 (3.6)	
SIMV	17 (7.4)	24 (10.7)		11 (17.2)	19 (8.6)	
APRV	6 (2.6)	7 (3.1)		0 (0.0)	9 (4.1)	
Intellivent-ASV	11 (4.8)	6 (2.7)		3 (4.7)	5 (2.3)	
other	38 (16.6)	16 (7.1)		11 (17.2)	17 (7.7)	
Vt, mL/kg PBW (median, IQR)	6.2 (5.7–6.9)	6.2 (5.7, 6.8)	0.561	6.3 (5.9–7.1)	6.2 (5.7–7.0)	0.295
PEEP (median, IQR)	12.0 (10.0–15.0)	12.0 (10.0–15.0)	0.419	12.0 (10.0–14.0)	12.0 (10.0–14.8)	0.848
Ppeak (median, IQR)	25.5 (23.0–29.0)	27.0 (24.0–30.0)	0.022	27.0 (24.0–30.0)	28.0 (25.0–32.0)	0.106
Driving pressure (median, IQR)	13.3 (12.0–16.0)	15.0 (12.0–16.8)	0.059	14.0 (13.0–16.2)	15.7 (13.0–18.2)	0.030
Mechanical power (median, IQR)	17.0 (13.2–19.8)	16.4 (13.6–19.8)	0.861	17.7 (13.9–21.4)	18.1 (14.8–22.6)	0.418
Compliance (median, IQR)	31.8 (26.1–40.6)	29.5 (24.1–38.2)	0.120	32.3 (26.0–38.2)	28.7 (22.3–34.5)	0.039
Total respiratory rate (median, IQR)	20.0 (18.0–22.8)	20.0 (20.0–24.0)	0.136	20.0 (18.0–21.5)	21.0 (18.0–25.0)	0.014
FiO_2_ (median, IQR)	0.6 (0.5–0.8)	0.6 (0.5–0.8)	0.127	0.7 (0.6–0.8)	0.8 (0.7–1.0)	0.016
SpO_2_/FiO_2_ ratio (median, IQR)	158.3 (125.5–192.1)	153.3 (116.7–180.0)	0.022	129.0 (109.1–141.5)	115.1 (96.2–136.5)	0.012
End tidal CO_2_ mmHg (median, IQR)	4.8 (4.3–5.5)	4.8 (4.2–5.5)	0.889	4.7 (4.3–5.6)	4.9 (4.3–5.7)	0.291
NMBA	89 (38.4)	116 (51.8)	0.005	27 (42.2)	133 (59.6)	0.015
**Vital signs**						
Mean arterial pressure, mmHg (median, IQR)	82.5 (72.0–98.0)	86.0 (75.0–101.0)	0.106	87.0 (74.0–97.2)	88.5 (75.0–105.2)	0.386
Heart rate, beats per min (median, IQR)	89.0 (77.0–105.5)	91.0 (78.0–105.0)	0.380	88.5 (76.5–100.2)	96.0 (83.0–112.0)	0.002
**Laboratory tests**						
pH (median, IQR)	7.4 (7.3–7.4)	7.4 (7.3–7.4)	0.083	7.4 (7.3–7.4)	7.3 (7.3–7.4)	0.224
PaO_2_ (median, IQR)	12.3 (10.1–15.6)	11.2 (9.5–13.8)	0.001	10.4 (9.3–11.8)	10.1 (8.9–11.4)	0.173
PaO_2_/FiO_2_ ratio (median, IQR)	161.5 (118.8–212.9)	143.8 (106.5–181.3)	0.005	108.8 (92.2–132.0)	96.4 (79.2–119.2)	0.005
PaCO_2_ (median, IQR)	5.5 (4.7–6.1)	5.6 (4.9–6.5)	0.037	5.9 (5.0–6.7)	6.2 (5.3–7.3)	0.031
Lactate (median, IQR)	1.2 (0.9–1.5)	1.2 (0.9–1.5)	0.713	1.2 (1.0–1.6)	1.2 (0.9–1.5)	0.466
Creatinine, µmol/L (median, IQR)	72.0 (58.0–91.0)	74.0 (62.0–100.0)	0.111	80.0 (68.0–96.0)	74.0 (57.0–91.2)	0.086

Abbreviations: SIMV: Synchronized intermittent mandatory ventilation, ASV: Adaptive Support Ventilation, APRV: Airway pressure release ventilation, PBW: predicted bodyweight, NMBA: neuromuscular blocking agents, Vt, Tidal Volume.

## Data Availability

Access to source data will be made available through local, regional and national anonymized datasets on request, and after agreement of the PRoVENT-COVID steering committee.

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
