# Peer review of "Incidence and Practice of Early Prone Positioning in Invasively Ventilated COVID-19 Patients—Insights from the PRoVENT-COVID Observational Study"

_jcm, 2021, doi:10.3390/jcm10204783_

Round 1
Reviewer 1 Report
Dear editors,
Thank you for the opportunity to review the submitted manuscript by Stilma and colleagues titled “Incidence and Practice of Prone Positioning in Invasively Ventilated COVID–19 Patients––insights from the PRoVENT– 3 COVID study”. In this study, the authors examine a prospective cohort of invasively ventilated COVID-19 patients, and examine 1) the incidence of prone positioning(PP), 2)patient factors associated with PP use and 3) associations between the use of PP and length of stay and survival outcomes.
I offer the following comments with the objective of helping improve the manuscript:
MAJOR comments
- I have significant concerns that this study is at risk for significant misclassification bias as described in the methods.Patients were allocated to the “indication for proning” group if their PF ratio and FiO2 met criteria as per the PROSEVA trial for two timepoints within the first 32 hours of invasive mechanical ventilation. Data on the use of prone positioning was collected up to four days after invasive mechanical ventilation. Many patients who present to ICU with COVID initially may be mild-moderate in severity and then subsequently develop severe. Specifically in this study if patients develop severe ARDS after the 32 hour mark, they may be proned but misclassified into the no indication group. This significantly impairs the ability of the reader to interpret the results. The high incidence of proning in the no-indication group may be the result of thos.
- The study as structured is at risk for Immortal Time bias.Patients who have an indication for prone positioning but die before receiving the treatment are automatically assigned to the control group. Also patients who are in the prone positioning group are immortal from the time of invasive ventilation to the time of the first prone positioning session. This should be addressed using an appropriate statistical method such as a time-varying cox regression analysis.
- How were appropriate exclusions to prone positioning assessed?Patients may be appropriately not considered for prone positioning. PROSEVA had a long list of reasons to exclude patients from prone positioning. Including patient in the indication group when they had a contra-indication would also lead to misclassification bias.
- NMBA use during prone positioning was much lower than that seen in PROSEVA.Please comment on how this may influence results.
- How many patients had missing data points?How was missing data handled in the analysis?
- How were chest-xrays and CT scans coded?For example the numbers of quadrants of airspace disease are noted. Was this based on the investigator’s interpretation or a radiologist? What was the relationship of the timing of the imaging to when the was deemed to have an indication for proning.
- ICU length of stay as an endpoint is difficult to intepret as it is dependant on patient’s being alive.LOS may decrease based upon an increase in patient death. The authors report LOS in survivors however please consider reporting 28 day vent free days as an alternative.
- Please comment on why delayed mortality (90 day) was lower in the proning group but 28 day mortality was not different between the groups.
MINOR comments
- The use of late prone positioning was not assessed (after 4 days) and may be a confounder. This should be discussed as a limitation
- The study should be reported according to the STROBE guidelines
Author Response
Thank for the valuable suggestions. A point to point reply is attached in the word document

Reviewer 2 Report
The authors report here data on the use of prone position in Covid-19 patients with ARDS. Data are original since they separated patients with and without a “classic” indication of prone position.
My comments are as follows:
-Introduction: Please refer first to the multicenter trial by Guérin (NEJM 2013) which showed a benefit from prone position in patients with ARDS and severe hypoxemia; and cite the meta-analysis by Sud (CMAJ 2014). The reference [1] is not appropriate.
-Paragraph 2.5. authors need to reference the indication they defined for prone positioning: reference Guérin NEJM 2013 ?
-The originality of this study is to have separate patients with and without a “classic” indication of prone position. Therefore to my opinion it would be better, rather than comparing the 4 groups together, to make statistical comparison (table 1 and 3) no prone vs prone in the group without indication on one hand, and no prone vs prone in the group with indication other hand. Indeed it seems (?) that patients without indication who were turned prone were the most severe in terms of oxygenation and respiratory mechanics (driving pressure and compliance); this point merits to be clarified.
-The analysis of data on mortality is quite confusing. The sentence appearing in the abstract and in the discussion “In patients with an indication for prone positioning mortality was lower in patients that received prone positioning than in patients that were left in the supine position” is not appropriate since the difference was not significant; please delete. Differences in crude mortality were not significant. Adjusted HRs were statistically different at day 90 among the four groups, but not when focusing on the group with indication (indication no prone vs indication prone). What is the message? Please clarify.
-Page 12 line 247 : reference 2 and not 20 ?
-Reference 10 is not correct : author is COVID-ICU group on behalf of the REVA Network and the COVID-ICU Investigators.
Author Response
Thank you for the valuable suggestions. A point to point reply is attached in the word document.

Round 2
Reviewer 1 Report
The authors have substantively and appropriately addressed the comments from the review. I have no further questions or concerns.
Reviewer 2 Report
The authors have responded to all my comments on the previous version, and the modifications provided to the manuscript are adequate.